# Valorization of Glycerol through the Enzymatic Synthesis of Acylglycerides with High Nutritional Value

**Daniel Alberto Sánchez** [1,3,*] , **Gabriela Marta Tonetto** [1,3] **and María Luján Ferreira** [2,3]

1   Departamento de Ingeniería Química, Universidad Nacional del Sur (UNS), Bahía Blanca 8000, Argentina; gtonetto@plapiqui.edu.ar
2   Departamento de Química, Universidad Nacional del Sur (UNS), Bahía Blanca 8000, Argentina; mlferreira@plapiqui.edu.ar
3   Planta Piloto de Ingeniería Química–PLAPIQUI (UNS-CONICET), Bahía Blanca 8000, Argentina
*   Correspondence: dsanchez@plapiqui.edu.ar; Tel.: +54-291-4861700

**Abstract:** The production of specific acylglycerides from the selective esterification of glycerol is an attractive alternative for the valorization of this by-product of the biodiesel industry. In this way, products with high added value are generated, increasing the profitability of the overall process and reducing an associated environmental threat. In this work, nutritional and medically interesting glycerides were obtained by enzymatic esterification through a two-stage process. In the first stage, 1,3-dicaprin was obtained by the regioselective esterification of glycerol and capric acid mediated by the commercial biocatalyst Lipozyme RM IM. Under optimal reaction conditions, 73% conversion of fatty acids and 76% selectivity to 1,3-dicaprin was achieved. A new model to explain the participation of lipase in the acyl migration reaction is presented. It evaluates the conditions in the microenvironment of the active site of the enzyme during the formation of the tetrahedral intermediate. In the second stage, the esterification of the sn-2 position of 1,3-dicaprin with palmitic acid was performed using the lipase from *Burkholderia cepacia* immobilized on chitosan as the biocatalyst. A biocatalyst containing 3 wt % of lipase showed good activity to esterify the sn-2 position of 1,3-dicaprin. A mixture of acylglycerides consisting mainly of capric acid esterified at sn-1 and sn-3, and of palmitic acid at the sn-2 position was obtained as the reaction product. The influence of the biocatalyst mass, the reaction temperature, and the molar ratio of substrates were evaluated for this reaction using a factorial design. Simple models were used to adjust the consumption of reagents and the generation of different products. The reaction product contained between 76% and 90% of acylglycerides with high nutritional value, depending on the reaction conditions.

**Keywords:** glycerol valorization; enzymatic esterification; nutritionally valuable acylglycerides; fat substitutes

## 1. Introduction

Glycerol (propane-1,2,3-triol) is an alcohol with a propane structure substituted in positions 1, 2, and 3 with hydroxyl groups. Glycerol is one of the main products of lipid digestion and an intermediate product of alcoholic fermentation. Furthermore, glycerol is the most important byproduct of the biodiesel industry and represents 10% by weight with respect to the biofuel produced [1]. Due to the global growth in the use of biofuels, the saturation of the market for this alcohol has taken place, and world production of three million tons was estimated for the year 2020, while the demand will be only 500,000 tons [2].

Purified glycerol can be used as an additive in the production of animal feed [3], in the wood industry [4], and can be transformed into a large number of products with high added value such as oxalic acid [5], propanediol [6,7], propanol [8], acrylic acid [9], polymers [10], and acylglycerides [11], among others.

Acylglycerides, also known as glycerides, are esters generated from glycerol and fatty acids (FA). In the glycerol backbone, one, two, or three of the hydroxyl groups may be esterified with fatty acids to form monoglycerides (MAG), diglycerides (DAG), and triglycerides (TAG), respectively. The nutritional value of acylglycerols is directly related to both the type of fatty acids that compose them and the position in which these acids are esterified in the glycerol molecule [12]. Glycerides with a specific structure are produced for nutritional or medical purposes [13]. In this way, medium-chain triacylglycerols (MCT) were used in child care and in infants with malabsorption problems due to their rapid digestion [14,15]. MCTs are absorbed more easily than long-chain triglycerides (LCT). MCTs are not toxic if their consumption is less than 30 g per day [16]. In addition, triglycerides that contain medium-chain fatty acids (MCFA) and long-chain fatty acids (LCFA) in the same molecule, called medium-long-chain triglycerides (MLCT), increase dietary-induced caloric expenditure, accelerate the energy production and reduce the accumulation of body fat. In addition, the use of MLCT could prevent obesity disorders and metabolic syndrome [17–20]. These triglycerides have been obtained by enzymatic acidolysis, esterification or interesterification [21–26].

MLCT with specific structure consist of MCFA at sn-1 and sn-3 positions, and LCFA at sn-2. They are called MLM-type triacylglycerols. These triglycerides have received special attention in recent years [27–31]. They have shown to have metabolic benefits over natural triglycerides, physical mixtures of triglycerides with random structure [32,33]. Particularly in infant nutrition, triglycerides with palmitic acid at the sn-2 position are better adsorbed than those esterified with C16 at sn-1 or sn-3 positions [34,35]. A variety of methods for the enzymatic synthesis of these compounds have been described [36,37]. These methods include the acidolysis of triglycerides [38,39] and interesterification between two triglycerides [40,41]. Unfortunately, the yields of MLM-type TAG produced by one-step synthesis are low, and a large number of unwanted by-products that are difficult to separate from the desired product are generated. One simple route to obtain MLM triglycerides is first to synthesize 1,3-diglycerides with 1,3-specific lipases, and then to esterify the desired fatty acid at the sn-2 position of glycerol (two-step synthesis). However, there are few reports in the literature about the synthesis of MLM-type triglycerides obtained from diglycerides [42–44], and the reported yields were considerably low.

On the other hand, diacylglycerols (DAG) are compounds present in a low proportion in oils and fats. Normally, the concentration of diglycerides in edible oils is below 5% [45,46]. Studies on the properties of diglycerides (especially 1,3-DAG) [47–50] indicate that they could reduce the concentration of triglycerides in serum [49], decrease body weight and visceral fat [47].

In this work, the valorization of glycerol was performed by the synthesis of nutritionally interesting acylglycerides in a two-stage enzymatic process. In the first stage, 1,3-dicaprin was obtained by esterification of glycerol and capric acid. Then, the dicaprin was esterified with palmitic acid. A mixture of acylglycerols containing mainly capric acid at the sn-1 and sn-3 positions and palmitic acid at the sn-2 position was obtained.

## 2. Results and Discussion

### 2.1. Synthesis of 1,3-Dicaproylglycerol

Since *Rhizomucor miehei* lipase (RML) is recognized as 1,3-specific, the theoretically expected products of glycerol and capric acid esterification are monocaprin (CGG) and the desired product: 1,3-dicaprin (CGC). Furthermore, a secondary reaction (acyl migration) generated undesired isomers of dicaprin: 1,2-dicaproylglycerol (CCG) and 2,3-dicaproylglycerol (GCC). Subsequent esterification of unwanted diglycerides generated tricaprin (CCC).

The identification of position isomers was carried out following the work of Bruschweiler and Dieffenbacher [51] and according to the separation capacity of the column used.

Table 1 shows the factors evaluated and the responses obtained for the esterification of glycerol and capric acid catalyzed by Lipozyme RM IM at six hours of reaction. The experimental factors were the glycerol mass (G), biocatalyst loading (B), and temperature (T).

**Table 1.** Enzymatic synthesis of 1,3-dicaproylglycerol, experimental factors, and responses obtained at 6 hours of reaction.

| Experiment Number | Experimental Factors | | | Response Variables | | | | |
|---|---|---|---|---|---|---|---|---|
| | G (mg) | B (mg) | T (°C) | $X_C$ (%) | $\sigma_{DAG}$ (%) | CCG | CGC | GCC |
| 1 | 50 | 20 | 40 | 38 | 74 | 5 | 92 | 3 |
| 2 | 150 | 30 | 50 | 66 | 70 | 4 | 92 | 4 |
| 3 | 250 | 20 | 40 | 55 | 83 | 10 | 83 | 7 |
| 4 | 250 | 40 | 40 | 67 | 76 | 17 | 80 | 4 |
| 5 | 250 | 40 | 60 | 75 | 71 | 9 | 71 | 20 |
| 6 | 150 | 30 | 50 | 67 | 72 | 5 | 94 | 1 |
| 7 | 250 | 20 | 60 | 73 | 76 | 5 | 93 | 2 |
| 8 | 50 | 20 | 60 | 45 | 58 | 8 | 86 | 5 |
| 9 | 50 | 40 | 60 | 55 | 33 | 3 | 33 | 64 |
| 10 | 50 | 40 | 40 | 40 | 56 | 3 | 56 | 40 |

CCG = 1,2-dicaproylglycerol, CGC = 1,3-dicaproylglycerol, GCC = 2,3-dicaproylglycerol.

The increase of all the variables studied generated an increase in the conversion of capric acid ($X_C$). In this study, biocatalyst aggregation was not detected, and lipase denaturation was not observed in the selected temperature range. The maximum conversion was obtained at the highest value of each experimental factor. However, the selectivity to dicaprin ($\sigma_{DAG}$) was not affected in the same way by these variables. The increase in glycerol concentration had a positive effect on the selectivity to dicaprin, while the increase in the amount of biocatalyst and temperature negatively affected this response. It is likely that a high initial concentration of glycerol promotes the generation of monocaprin and subsequently 1,3-dicaprin. On the other hand, the increase in temperature and mass of the biocatalyst favors the formation of 1,2/2,3-dicaprin (due to the acyl migration reaction) and then these diglycerides are re-esterified to generate tricaprin. The negative effects of the increase in the biocatalyst load and the temperature on the specific diglyceride selectivity became important when the glycerol concentration was low.

Although RML is recognized as 1,3-specific, diglycerides with capric acid at the sn-2 position were obtained: 1,2-dicaprin and 2,3-dicaprin. The generation of different isomers was influenced by a combination of factors. In general terms, the synthesis of non-specific isomers was positively influenced by the increase in the biocatalyst load and the reaction temperature. On the other hand, carrying out the reaction with high initial glycerol concentration favored the generation of the desired diglyceride (1,3-dicaproylglycerol). The factors that improved the obtaining of unwanted diglycerides also favored their subsequent esterification and the generation of tricaprin.

The generation of 1,2-dicaprin was mainly influenced by the immobilized lipase load and the initial concentration of glycerol absorbed in silica gel. The high glycerol content reduced the formation of this isomer, while a high load of Lipozyme RM IM promoted the synthesis of 1,2-dicaproylglycerol. When carrying out the reaction with high nominal glycerol concentration the effect of temperature was not significant. However, at low concentrations of glycerol, the fraction of 1,2-dicaprin detected at 6 h of reaction was reduced as a result of the increase in reaction temperature. Apparently, raising the reaction temperature favors the esterification of 1,2-dicaprin with capric acid and the generation of tricaprin.

The generation of 2,3-dicaprin is also related to the acyl migration reaction in 1,3-dicaprin. The effect of the variables in this response was similar to that described for 1,2-dicaprin. In summary, the synthesis of undesired isomers was favored by the increase in the mass of immobilized lipase used, in particular at low initial glycerol concentrations. These results allow us to presume the participation of the catalyst in the acyl migration reaction. The effect of the biocatalyst load on the secondary reaction was even greater than that produced by the reaction temperature.

Contrary to what was observed for unwanted diglycerides, the generation of 1,3-dicaprin was negatively affected by the increase in the loading of Lipozyme RM IM, indicating the possible participation of lipase in the secondary reaction. The increase in temperature had a slight positive effect on the synthesis of the desired diglyceride, probably the temperature favored the esterification reaction more than the acyl migration. However, increasing the reaction temperature at low glycerol concentrations results in a reduction in the concentration of 1,3-dicaprin. It is likely that the low nominal concentration of glycerol generates a high concentration of 1,3-diglyceride in the environment of the active site of the lipase, promoting the coordination of the diglyceride, and the acyl migration reaction.

The optimal conditions for the synthesis of 1,3-dicaprin are highlighted in Table 1.

Acyl Migration Promoted for Lipase

Several authors have reported the increase in acyl migration associated with the increase in lipase content and reaction temperature. However, no studies were presented with the objective of exploring this topic in depth [52,53]. In previous work, we studied the participation of Lipozyme RM IM support in the acyl migration reaction [54]. The results showed the lack of participation of this solid in the acyl migration reaction towards the sn-2 position. This makes us assume that lipase could catalyze isomerization by acylation migration and this reaction could be related to the conditions of the environment of the active site of the lipase.

Spontaneous isomerization from 1,2-DAG to 1,3-DAG and from 2-MAG to 1-MAG has been described by Laszlo et al. [55]. The formation of an intermediary, called the ketal intermediary, was proposed to explain the migration from 1,2-DAG to 1,3-DAG. This mechanism has been proposed for the liquid state and the formation of this intermediate could be catalyzed by acidic or basic groups, or it could even occur in an uncatalyzed form in the reaction medium. A similar intermediary is generated during the coordination of the lipase with the diglyceride. In this case, we should consider the structure of the active site and the location of the compound in its vicinity. The structure of the *Rhizomucor miehei* lipase (RML) was obtained from Protein Data Bank 3tlg [56]. This lipase has in its catalytic triad a Serine residue at position 144, Histidine at position 257, and Aspartate at position 203, spatially located in a specific way. Figure 1 shows the coordination of 1,3-dicaproylglycerol with the Serine residue of the catalytic triad.

In Figure 1, the migration and the new possible bonds are shown with a red dotted line. In both cases, the DAG was coordinated to the Serine residue and the intermediary was formed. In the schemes, the coordination of 1,3-dicaprin occurs through carbonyl at the sn-1 position. However, there are two possibilities for acyl migration. Figure 1I shows the migration of the acyl initially located at the sn-1 position, the product of this migration is shown in Figure 2a.

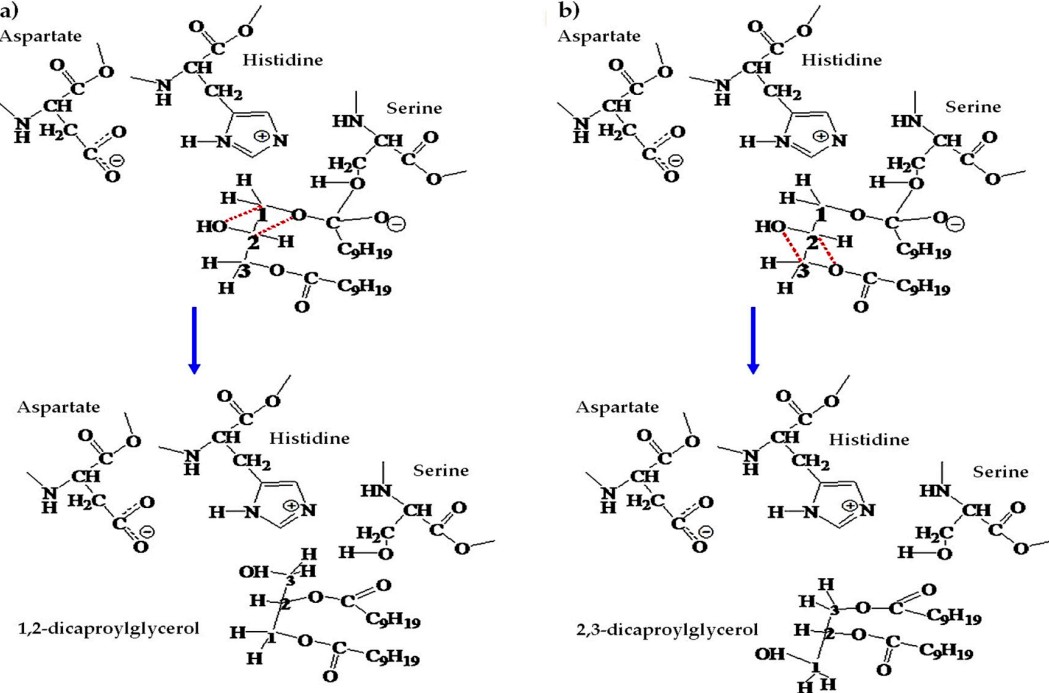

**Figure 1.** Scheme of 1,3-DAG with OH of C2 positioned near Histidine. (**I**) Acyl migration with the participation of the Serine and (**II**) Acyl migration mediated by the distribution of charges around the catalytic triad.

**Figure 2.** Acyl migration in 1,3-dicaprin. Coordination through position sn-1, (**a**) formation of 1,2-dicaprin by migration mediated by the Serine residue and (**b**) generation of 2,3-dicaprin by migration mediated by the distribution of charges from the environment of the active site.

The sn-1 and sn-3 positions of 1,3-dicaprin are indistinguishable in solution, and the carbon at sn-2 is non-chiral. When coordination with Serine occurs and the tetrahedral intermediate is formed, C2

becomes chiral. The intermediate formed is different depending on the position of the acyl coordinated with the Serine residue. In Figures 1 and 2, the intermediary was generated with the OH of C2 towards the side of the Histidine residue. In order to observe the differences in the tetrahedral intermediate, in Figure 3 the enzyme-mediated acyl migration reaction is schematized when the OH of C2 is close to the Serine residue.

**Figure 3.** Diagram of acyl migration in 1,3 DAG with the OH of C2 positioned towards the Serine side.

Depending on what was observed in Figures 1–3, coordination by C1 (Figures 1 and 2) and acyl migration seems simpler than in the case of interaction by C3 (Figure 3), where the acyl migration would be less favored due to steric hindrances. This steric hindrance is clearly seen in Figure 4. A simplified three-dimensional view of the coordination of the Serine residue with 1,3-diacetin (as a model) is shown. The coordination of the Serine with the acyl at the sn-3 position is schematized in Figure 4a. The steric impediment to migration is greater than in Figure 4b. This figure shows the coordination of the Serine with the acyl at the sn-1 position, a situation in which the acyl migration seems more likely.

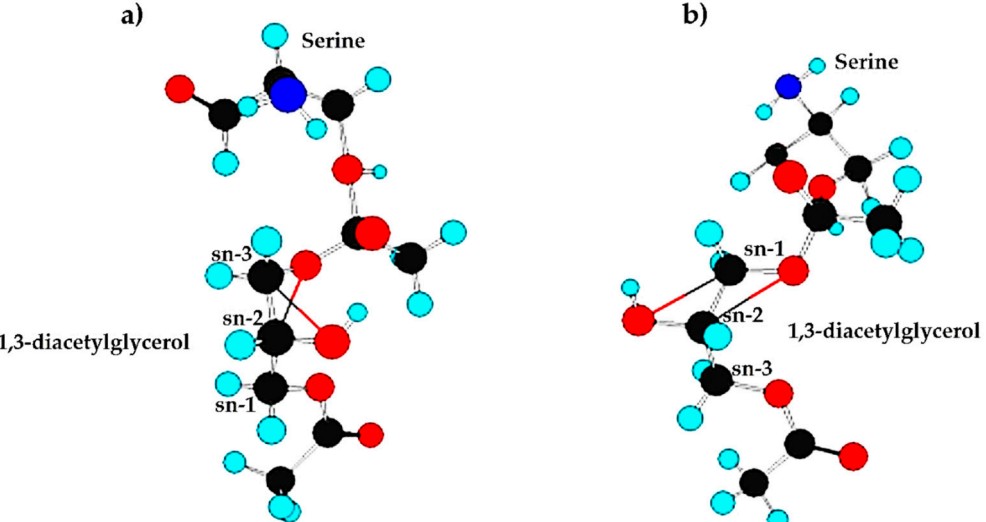

**Figure 4.** Three-dimensional representation of the coordination of 1,3-diacetin with the Serine residue of the catalytic triad. (**a**) Interaction through acyl at the sn-3 position and (**b**) interaction through acyl at the sn-1 position.

Based on the analysis of the steric environment of the active site of RML, it is clear that the acyl migration in 1,3-dicaprin promoted by the Serine residue is more likely when coordination occurs through the sn-1 position. The Serine of the catalytic triad and the distribution of charges in the environment of the active site could participate in the acyl migration. When the participation of the Serine residue in the isomerization of 1,3-dicaprin is considered, the product with the highest probability of formation is 1,2-dicaprin.

Experimental results showed that the synthesis of unwanted isomers was favored by the increase of the lipase mass and reaction temperature, as expected if the isomerization is kinetically controlled. Acyl migration was higher at low glycerol concentrations, under these conditions a relatively high concentration of 1,3-dicaprin is close to the active lipase site, favoring its coordination with Serine, the formation of a tetrahedral intermediate and subsequent acyl migration.

The proposal herein seeks to study how the environment of the active site of the enzyme can generate the conditions to explain, although partially, the experimental results. This is a novel model that would explain the participation of the enzyme in the acyl migration reaction, it is based on experimental results [54,57] and is consistent with theoretical–experimental results obtained for the acyl migration reaction in uncatalyzed form [55].

### 2.2. Immobilization of Lipase from Burkholderia on Chitosan

*Burkholderia cepacia* lipase (BCL) was immobilized on chitosan flakes (obtained from prawn shells with a degree of deacetylation of 85.2%) following a previously published methodology [12]. The molecular weight of this material was between 70,000 and 80,000 g/mol and the BET surface area was between 3 and 5 m$^2$/g. Figure 5 shows two scanning electron microphotographs of the chitosan.

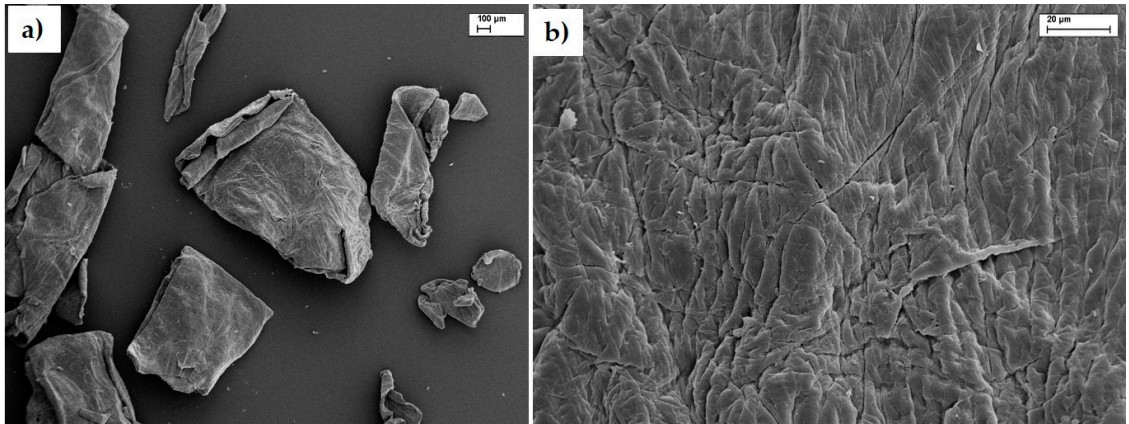

**Figure 5.** Scanning electron microphotographs of chitosan flakes: (**a**) 61×, (**b**) 1500×.

The determination of the initial content of lipase and the remaining lipase in the supernatant after immobilization was carried out by quantification of sulfur by ICP-AES [12,58]. The immobilization process began with 99.97 ± 1.24 mg of lipase and 67.92 ± 1.95 mg of lipase was recovered after the immobilization process and the solid washes. Thus, 32.05 ± 3.19 mg of lipase were immobilized on 1 g of chitosan. This procedure had an immobilization efficiency close to 32% and a biocatalyst containing 3.1 wt % of lipase was obtained. This catalyst was denoted as BCL/chitosan.

### 2.3. Esterification of Dicaprin and Palmitic Acid Catalyzed by BCL/Chitosan

1,3-Dicaproyl-2-palmitoyl glycerol (CPC) was obtained as the product of interest in the esterification 1,3-dicaprin (CGC) and palmitic acid (P). Furthermore, esterification of the 1,2-dicaprin fraction present in the starting material generated 1,2-dicaproyl-3-palmitoylglycerol (CCP).

The biocatalyst was thermally treated to remove the weakly adsorbed water, but the hydrolysis could not be avoided. The generation of capric acid (C) and glycerol (G) was indicative of the occurrence of this reaction. The products of the hydrolysis and subsequent esterification also included monopalmitin (PGG), 1-caproyl-2-palmitoyl glycerol (CPG), 1-caproyl-3-palmitoyl glycerol (CGP), tricaprin (CCC), dipalmitin (PGP), and 1-caproyl-2,3-dipalmitoyl glycerol (CPP).

In previous studies [54,57], we reported the ability of the capillary column used in this work to identify isomers as a function of the position of the hydroxyl group. Thus, it was possible to identify and quantify isomers of mono- and diglycerides, but not triglyceride isomers. To identify the triglyceride isomers, it was necessary to perform a regioselective hydrolysis with porcine pancreas lipase.

A chromatogram of the reaction products and reagents not consumed in the esterification mediated by BCL/chitosan is shown in Figure 6.

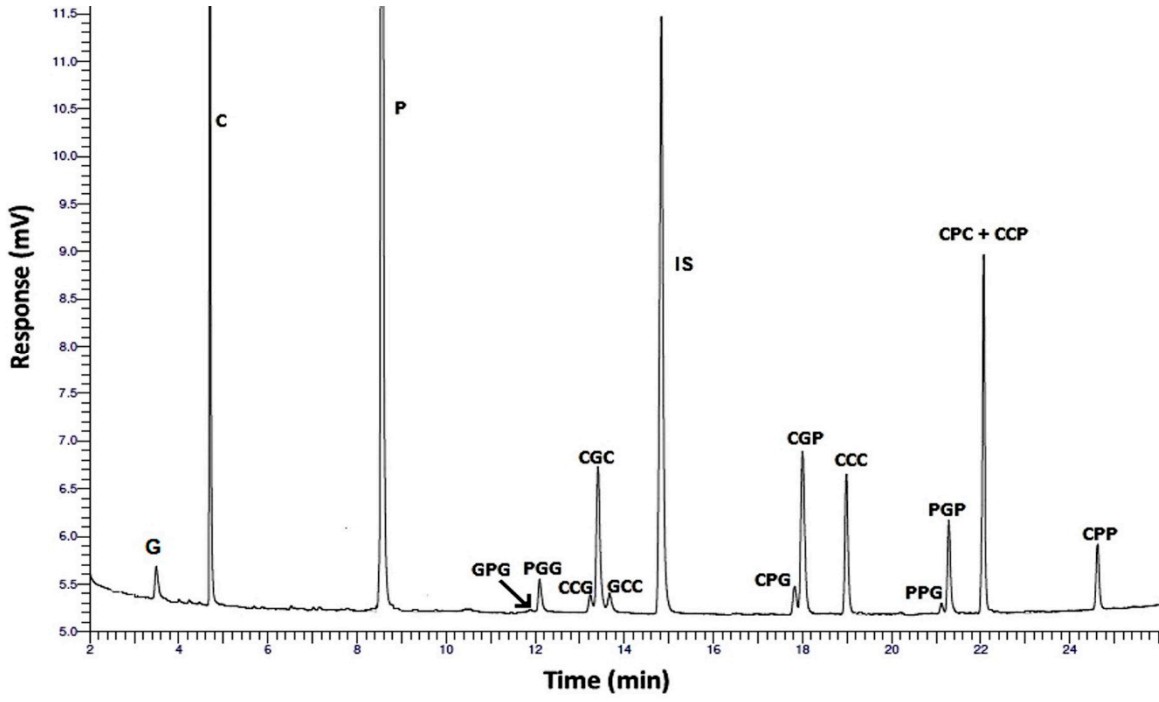

**Figure 6.** Chromatogram of a reaction sample obtained at 6 h reaction during the esterification of dicaprin and palmitic acid. The experimental conditions were as follows: dicaprin = 32 mg, $R_M$ = 3 mol/mol, temperature = 40 °C and biocatalyst load = 150 mg. IS: Internal calibration standard.

### 2.3.1. Model Fitting and ANOVA

The results presented in Tables 2–4 correspond to measurements made at 6 h of reaction. The results for each response are analyzed in the following sections and they were adjusted using second-order models. In these models, the Fisher–Snedecor test (F-test) [59] was applied to eliminate statistically non-significant variables.

**Table 2.** Established experimental factors and reagents consumed during enzymatic esterification of dicaprin with palmitic acid catalyzed by BCL/chitosan at 6 h of reaction.

| Experiment Number | Experimental Factors | | | Reagents Consumed (µmoles) | | |
|:---:|:---:|:---:|:---:|:---:|:---:|:---:|
| | B (mg) | T (°C) | $R_M$ (mol/mol) | CCG | CGC | P |
| 1 | 150 | 60 | 1 | 11.2 | 57.3 | 40.0 |
| 2 | 50 | 60 | 3 | 11.6 | 43.2 | 37.7 |
| 3 | 50 | 60 | 1 | 10.4 | 38.5 | 31.3 |
| 4 | 150 | 40 | 1 | 10.3 | 52.0 | 47.5 |
| 5 | 50 | 40 | 3 | 11.5 | 31.9 | 38.2 |
| 6 | 150 | 60 | 3 | 12.4 | 57.1 | 62.9 |
| 7 | 100 | 50 | 2 | 11.7 | 51.9 | 50.6 |
| 8 | 150 | 40 | 3 | 12.0 | 53.4 | 66.3 |
| 9 | 100 | 50 | 2 | 11.9 | 53.3 | 49.8 |
| 10 | 50 | 40 | 1 | 11.5 | 34.6 | 35.1 |

**Table 3.** Experimental factors and products generated during the enzymatic esterification of dicaprin with palmitic acid catalyzed by BCL/chitosan at 6 h of reaction.

| Experiment Number | Experimental Factors | | | Products Generated (µmoles) | | | | | | | |
|---|---|---|---|---|---|---|---|---|---|---|---|
| | B (mg) | T (°C) | $R_M$ (mol/mol) | GGG | C | PGG + GPG | CPG + CGP | CCC | PGP + PPG | CPC + CCP | CPP |
| 1 | 150 | 60 | 1 | 25.3 | 65.3 | 1.9 | 10.5 | 6.7 | 2.5 | 17.7 | 2.5 |
| 2 | 50 | 60 | 3 | 15.9 | 50.7 | 2.5 | 12.6 | 3.6 | 1.6 | 17.2 | 1.1 |
| 3 | 50 | 60 | 1 | 8.9 | 24.9 | 2.2 | 10.2 | 11.0 | 1.2 | 15.0 | 0.7 |
| 4 | 150 | 40 | 1 | 16.7 | 56.2 | 2.4 | 13.4 | 3.7 | 3.1 | 20.7 | 2.4 |
| 5 | 50 | 40 | 3 | 1.4 | 19.3 | 2.4 | 13.4 | 6.7 | 1.7 | 16.2 | 1.4 |
| 6 | 150 | 60 | 3 | 11.3 | 53.8 | 2.9 | 14.7 | 7.2 | 5.8 | 21.3 | 6.2 |
| 7 | 100 | 50 | 2 | 18.1 | 68.4 | 2.9 | 16.2 | 1.3 | 3.9 | 17.8 | 2.9 |
| 8 | 150 | 40 | 3 | 7.0 | 49.5 | 2.6 | 19.1 | 3.3 | 5.7 | 22.8 | 4.7 |
| 9 | 100 | 50 | 2 | 20.8 | 71.6 | 3.0 | 17.0 | 1.2 | 3.8 | 17.0 | 2.7 |
| 10 | 50 | 40 | 1 | 6.1 | 25.6 | 2.3 | 13.4 | 8.3 | 1.9 | 14.1 | 0.6 |

**Table 4.** Experimental factors and parameters evaluated during the enzymatic esterification of dicaprin with palmitic acid catalyzed by BCL/chitosan at 6 h of reaction.

| Experiment Number | Experimental Factors | | | 1,3-Dicaprin Conversión (%) | Valuable Acylglycerides (%) [a] | Valuable Acylglycerides (%) [b] |
|---|---|---|---|---|---|---|
| | B (mg) | T (°C) | $R_M$ (mol/mol) | | | |
| 1 | 150 | 60 | 1 | 90 | 68.1 | 80.9 |
| 2 | 50 | 60 | 3 | 68 | 54.6 | 88.4 |
| 3 | 50 | 60 | 1 | 60 | 52.4 | 89.2 |
| 4 | 150 | 40 | 1 | 81 | 62.1 | 81.6 |
| 5 | 50 | 40 | 3 | 50 | 47.8 | 90.0 |
| 6 | 150 | 60 | 3 | 89 | 65.9 | 76.2 |
| 7 | 100 | 50 | 2 | 81 | 60.5 | 81.0 |
| 8 | 150 | 40 | 3 | 84 | 64.5 | 79.5 |
| 9 | 100 | 50 | 2 | 83 | 62.4 | 81.2 |
| 10 | 50 | 40 | 1 | 54 | 49.7 | 90.3 |

[a] Acylglycerides with high nutritional value without considering the presence of 1,3-dicaprin in the final product. [b] Acylglycerides with high nutritional value taking into account unreacted 1,3-dicaprin in the final product.

**Table 5.** Results of the models obtained with the Statgraphics Centurion XV software for each of the responses studied in the dicaprin and palmitic acid esterification catalyzed by BCL/chitosan. All data had a confidence level greater than 95.0%.

| Response | Equation | Eq. n | $R^2$ (%) | *p*-Value | F-Value |
|---|---|---|---|---|---|
| 1,3-dicaprin conversion | $X_{CGC} = -9.75 + 1.08B + 0.475T - 0.004B^2$ | (2) | 97.0 | 0.0001 | 64.15 |
| 1,2-dicarpin conversion | $X_{CCG} = 76.91 - 0.00114B^2 - 0.3141T + 0.00352TB + 0.0323BR_M$ | (3) | 88.1 | 0.0157 | 9.23 |
| palmitic acid conversion | $P = 6.23 - 0.00268B^2 + 0.682B - 0.00193BT + 0.680BR_M$ | (4) | 99.1 | 0.0000 | 140.8 |
| capric acid generation | $C = -24.8 + 0.833B - 6.26R_M^2 - 0.164R_MB + 0.862R_MT + 0.00672BT$ | (5) | 92.6 | 0.005 | 15.6 |
| glycerol generation | $GGG = -7.24 + 0.240B - 7.25R_M - 0.108BR_M + 0.325TR_M$ | (6) | 91.2 | 0.03 | 8.3 |
| monopalmitin generation | $Monopalmitin = 1.38 - 0.392R_M^2 + 1.40R_M - 0.000142BT + 0.00503BR_M$ | (7) | 96.4 | 0.0008 | 33.5 |
| CPG + CGP generation | $CPG + CGP = 1.77 - 0.00128B^2 + 0.310B - 0.00131BT + 0.0161BR_M$ | (8) | 95.9 | 0.0011 | 29.5 |
| dipalmitin generation | $Dipalmitin = 1.35 - 0.000368B^2 + 0.0719B - 0.05075T - 1.50R_M + 0.01426BR_M + 0.0170TR_M$ | (9) | 99.9 | 0.0001 | 667 |
| tricaprin generation | $CCC = 30.1 + 0.00202B^2 - 0.573B + 0.0216BR_M - 0.0648TR_M + 0.00207BT$ | (10) | 93.7 | 0.0163 | 11.9 |
| CPP generation | $CPP = -0.484 + 0.00970BR_M + 0.000213BT$ | (11) | 96.4 | 0.0000 | 94.1 |
| CPC + CCP generation | $CPC + CCP = 10.3 + 0.101B + 0.0263TR_M - 0.00101TB$ | (12) | 96.2 | 0.0001 | 50.4 |
| valuable glycerides [a] | $Glycerides = 23.61 + 0.405B + 0.211T - 0.00133B^2$ | (13) | 97.4 | 0.0000 | 75.19 |
| valuable glycerides [b] | $Glycerides = 103.8 - 0.0344B + 2.5R_M + 0.00137B^2 - 0.0143BR_M - 0.0413TR_M$ | (14) | 99.7 | 0.0000 | 237.65 |

[a] Acylglycerides with high nutritional value without considering the presence of 1,3-dicaprin in the final product. [b] Acylglycerides with high nutritional value taking into account unreacted 1,3-dicaprin in the final product.

Equation (1) represents the second-order model containing all the variables and combinations of them:

$$X_{Ac} = A_0 + A_1B + A_2T + A_3R_M + A_4BT + A_5BR_M + A_6TR_M + A_7B^2 + A_8T^2 + A_9R_M{}^2 \quad (1)$$

where $R_M$ is the P/dicaprin molar ratio, T is temperature, B is the amount of biocatalyst, and $A_i$ are the regression coefficients of the model (intercept, linear, interaction, and quadratic terms). Table 5 lists all the equations obtained after multiple regression using the Statgraphics Centurion XV.2 software (Equations (2)–(14)), the description of the responses, the equation numbers, the percentages of variation of the parameters explained using $R^2$, the *p*-values and the F-values.

Consumed Reagents

- 1,3-Dicaprin Conversion

As mentioned above, this diglyceride was consumed by two pathways. Firstly, the esterification at the sn-2 position allowed the generation of the structured triglyceride (CPC). Secondly, the hydrolysis reaction generated capric acid and glycerol. After 6 h of reaction, the presence of monocaprin was not detected.

The increase in the values of all the variables favored the conversion of 1,3-dicaprin ($X_{CGC}$), however the effect of the increase of $R_M$ was virtually insignificant on this response. The conversion values for 1,3-dicaprin were between 50% and 90%.

Equation (2) (obtained through multiple regression and subsequent refinement) accounted for 97.0% of the changes observed in the conversion of 1,3-dicaprin.

- 1,2-Dicaprin Conversion

Under the optimum conditions found for the synthesis of 1,3-dicaprin, the 1,3-dicaprin/1,2(2,3)-dicaprin ratio in the product was approximately 9:1. However, in the separation process [60], the lost diglyceride was mostly 1,3-dicaprin (verified by gas chromatography), and for this reason the 1,2-DAG fraction was higher at the beginning of the reaction (1,3-dicaprin/1,2-dicaprin ratio = 8:2).

Like its isomer, 1,2-dicaprin is consumed in two reactions: hydrolysis and esterification at the sn-3 position (which generates 1,2-dicaproyl-3-palmitoyl glycerol). The conversion of 1,2-dicaprin ($X_{CCG}$) was positively affected by the increase of $R_M$ and the temperature. This response was negatively affected by the increase of the biocatalyst mass. Probably, CCC and CCP are hydrolyzed and converted back to 1,2-dicaprin. The conversion values of 1,2-dicaprin were between 64% and 78%.

Equation (3) accounted for 88.1% of the changes detected in the conversion of this diglyceride (as a function of $R^2$).

- Palmitic Acid Conversion

In the reaction system under study, palmitic acid was consumed to give the theoretically expected products (1,3-dicaproyl-2-palmitoyl glycerol and in minor proportion 1,2-dicaproyl-3-palmitoyl glycerol), and it also reacted with the products of the hydrolysis reaction to give other compounds such as monopalmitin, dipalmitin, 1-caproyl-2-palmitoyl glycerol, 1-caproyl-3-palmitoyl glycerol and 1-caproyl-2,3-dipalmitoyl glycerol.

The molar ratio of palmitic acid to dicaprin ($R_M$) was varied between 1:1 and 3:1 in order to evaluate the influence of this parameter on the yield of the desired glycerides.

The increase in B and $R_M$ favored the conversion of palmitic acid. Apparently both factors had a positive impact on the esterification reaction. The increase in temperature showed a negative effect on the consumption of this acid during reaction. Hydrolysis is probably more favored by the increase in temperature than esterification under these conditions. Usually, hydrolysis has a lower activation energy than the esterification reaction.

Equation (4) was obtained after performing multiple regression, and it accounted for 99.1% of the observed changes in the micromoles of palmitic acid consumed after 6 h of reaction.

Products Generated by Hydrolysis

- Generation of Capric Acid

During the esterification of dicaprin with palmitic acid, a secondary (but not minor) reaction took place. Although the reaction system did not contain water, hydrolysis was important in this study, even though the weakly adsorbed water was removed from the biocatalyst. The water contained in the immobilized lipase, essential to preserve its activity, could allow the hydrolysis reaction to occur. In addition, esterification produces water as a by-product.

This undesired reaction can be evaluated as a function of the generation of capric acid.

The increase in temperature and biocatalyst mass had an effect in favor of hydrolysis reaction. However, it can be seen that high biocatalyst loads reduce the generation of capric acid. This effect may be related to the fact that a high biocatalyst load allows the re-esterification of free fatty acids. On the other hand, and as mentioned above, hydrolysis generally has a lower activation energy than the esterification reaction and the increase in temperature favors hydrolysis more than esterification. Finally, the increase in palmitic acid mass had a negative effect on the hydrolysis reaction. A higher concentration of acid in the reaction medium promotes a greater concentration of the same in the environment of the enzyme, displacing the water and favoring the esterification against hydrolysis. In the proportions of palmitic acid evaluated in this work, no inhibition of the lipase by this substrate was observed.

The surface response graphs presented in Figure 7 show the effect of the variables on the generation of capric acid (indirectly on the hydrolysis reaction). The plots were obtained from Equation (5), which adjusts this response with a $R^2 = 92.6\%$.

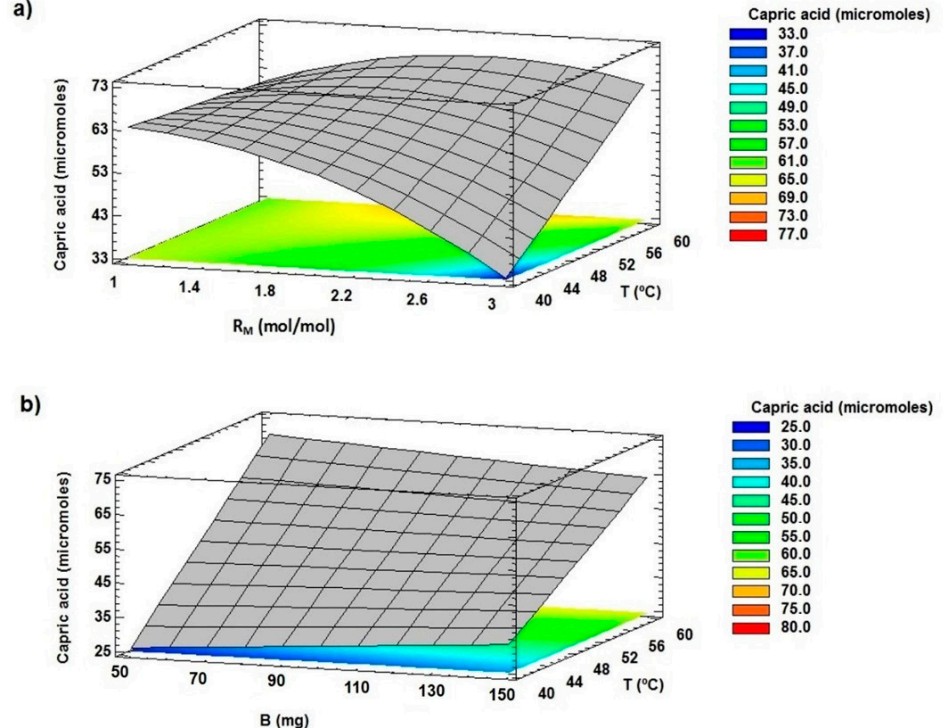

**Figure 7.** Capric acid generated by hydrolysis during the esterification of dicaprin with palmitic acid catalyzed by BCL/chitosan. (**a**) Variables: $R_M$ and reaction temperature for a 150 mg biocatalyst dosage, (**b**) Variables: biocatalyst mass and reaction temperature for a $R_M$ = 1:1 (mol/mol).

- Generation of Glycerol

The generation of glycerol is a clear indication of how important the hydrolysis reaction was. As in the generation of capric acid, the increase in temperature and immobilized lipase content favored the hydrolysis reaction, and in this case the synthesis of glycerol. However, when the initial palmitic acid content was high, glycerol formation did not increase with increasing biocatalyst dosage.

Equation (6), which was obtained by multiple regression and refined by removing the statistically non-significant variables, accounted for 91.2% of the changes in µmoles of generated glycerol (as a function of $R^2$).

Products Generated by Hydrolysis and Subsequent Esterification

- Generation of Monopalmitin

After the hydrolysis of dicaprin and the generation of glycerol, the latter was esterified with palmitic acid to generate monopalmitin, mainly 1-monopalmitin and to a lesser extent 2-monopalmitin (as determined by gas chromatography). Based on the objectives of the present study, both monoglycerides were regarded as undesirable and quantified together (Table 4).

The generation of these monoglycerides was positively affected by the increase in $R_M$ and B. The increase in temperature had a negative impact on the synthesis of monopalmitin. As mentioned above, this variable had a very important effect on hydrolysis, which would explain this behavior.

The relationship between this response and the studied variables is represented by Equation (7), with a coefficient of determination $R^2 = 96.4\%$.

- Generation of Diglycerides Formed by Capric and Palmitic Acid

The hydrolysis reaction, which occurred mainly at the sn-3 position, allowed the generation of 1-monocaprin. It is possible that the monoglyceride could have been rapidly esterified with palmitic acid at the sn-3 position giving 1-caproyl-3-palmitoyl glycerol (CGP), which would explain the absence of monocaprin after 6 h of reaction.

On the other hand, 1-caproyl-2-palmitoyl glycerol (CPG) could be generated by the esterification of 1-monocaprin at sn-2 position, or after the hydrolysis at sn-3 position of MLCT (1,3-dicaproyl-2-palmitoyl glycerol (CPC) and 1-caproyl-2,3-dipalmitoyl glycerol (CPP)).

Both diglycerides are interesting from a nutritional point of view, and for that reason they were quantified together. It is known that 1,3-diglycerides are easily metabolized and provide a rapid source of energy. On the other hand, 1,2-diglycerides containing a long-chain fatty acid at the sn-2 position and a medium-chain fatty acid at the sn-1/sn-3 position would have a similar behavior to that of MLM-type MLCT. The metabolism of these triglycerides leads to the formation of the diglycerides in question. These diglycerides will be denoted as medium- and long-chain diacylglycerols (MLCD).

The increase in $R_M$ and biocatalyst content favored the synthesis of these diglycerides. Both factors favored the esterification of P, in agreement with what was mentioned above. On the other hand, the increase in temperature had a negative effect on this response. It is again evident that high temperatures produce a significant increase in the rate of the hydrolysis reaction.

Figure 8 shows surface plots for this response obtained from Equation (8), which correlates the generation of the diglycerides with the evaluated factors with a $R^2 = 95.9\%$.

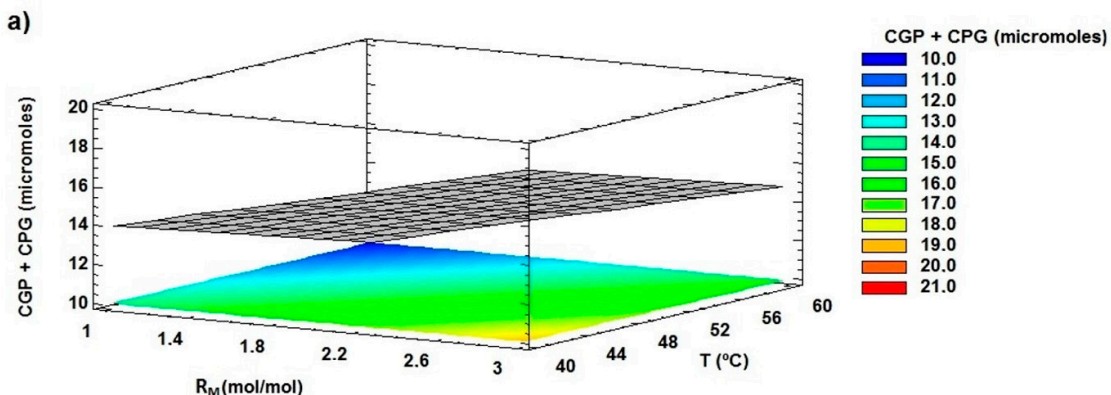

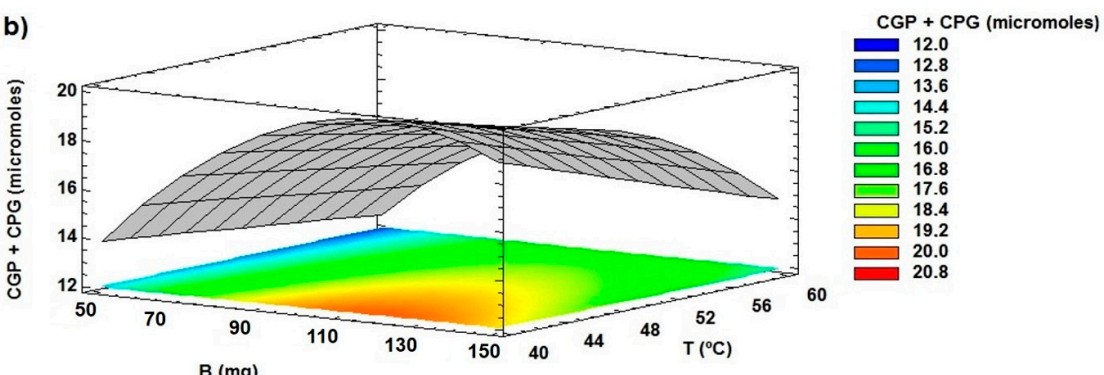

**Figure 8.** Generation of 1-caproyl-3-palmitoyl glycerol (CGP) + 1-caproyl-2-palmitoyl glycerol (CPG) during the esterification of dicaprin with palmitic acid catalyzed by chitosan-immobilized *Burkholderia cepacia* lipase. (**a**) Variables: $R_M$ and reaction temperature for a 150 mg biocatalyst dosage, (**b**) Variables: biocatalyst mass and reaction temperature for a $R_M$ = 3:1 (mol/mol).

- Generation of Dipalmitin

The formation of dipalmitin could be attributed to the esterification of glycerol (generated by the hydrolysis of dicaprin) with palmitic acid. As shown in Figure 5, 1,3-dipalmitin was the main product obtained, and a smaller proportion of 1,2-dipalmitin was also generated.

The increase in biocatalyst dosage and $R_M$ favored the generation of these diglycerides, while temperature affected negatively this response, although its effect was not significant except when the concentration of palmitic acid was low.

Equation (9) was obtained by multiple regression and subsequent refinement. The coefficient of determination for this model was 99.9%.

- Formation of Tricaprin

Medium chain-length triglycerides are nutritionally valuable, and therefore the presence of tricaprin in the reaction product increases its nutritional value. The capric acid obtained by hydrolysis is re-esterified to generate tricaprin.

The increase in the reaction temperature favored the generation of tricaprin, especially when the concentration of palmitic acid was low, indicating that a high $R_M$ would favor the esterification of palmitic acid. On the other hand, carrying out the synthesis with a low biocatalyst content had a negative impact on this response. It is probable that there is a competition between the hydrolysis and esterification reactions, where the biocatalyst dosage plays a significant role. Finally, the increase in palmitic acid concentration had a negative effect on the generation of tricaprin. As mentioned above, the increase in the values of this factor favored the esterification of palmitic acid instead of capric acid.

Equation (10) represents the model obtained by multiple linear regression to describe the relationship between the generated tricaprin in μmoles and the selected factors ($R^2$ for the obtained model = 93.7%).

- Generation of 1-Caproyl-2,3-Dipalmitoyl Glycerol

The diglycerides containing capric acid and palmitic acid in the same molecule were again esterified with palmitic acid. This reaction generated 1-caproyl-2,3-dipalmitoyl glycerol (CPP).

As in the case of monopalmitin and dipalmitin, the formation of this triglyceride should be minimized to increase the nutritional value of the final product.

All the analyzed factors had a positive effect on this response. This result was expected because the generation of this triglyceride depends on the hydrolysis reaction and the esterification of palmitic acid. A simple model represented by Equation (11) accounted for 96.4% of the changes in CPP generation.

Products Generated from the Esterification of Dicaprin with Palmitic Acid

- Formation of 1,3-Dicaproyl-2-Palmitoyl Glycerol and 1,2-Dicaproyl-3-Palmitoyl Glycerol

The esterification at the sn-2 position of 1,3-dicaprin with palmitic acid catalyzed by immobilized *Burkholderia cepacia* lipase was favored by the correct selection of the reaction solvent. The product of this reaction was 1,3-dicaproyl-2-palmitoyl glycerol, a triglyceride with high nutritional value.

Another nutritionally interesting MLCT (but less than CPC) is 1,2-dicaproyl-3-palmitoyl glycerol, which was obtained from the esterification of 1,2-dicaprin at the sn-3 position.

High concentrations of 2-monopalmitin and 1-caproyl-2-palmitoyl glycerol were detected after carrying out the hydrolysis of the reaction product using PPL. These results indicated that a high proportion of P was esterified at the sn-2 position. This result was corroborated using nuclear magnetic resonance (NMR) (results not shown).

Since the aim of the present work is to obtain a mixture of acylglycerides that are nutritionally attractive, both triglycerides were quantified together.

The increase in biocatalyst content and $R_M$ favored the synthesis of these triglycerides. On the other hand, the increase in temperature negatively affected this response. These results are in agreement with previously observed results: high relative initial palmitic acid content, high biocatalyst to substrate mass ratio and low temperature favored the esterification of palmitic acid versus hydrolysis.

Equation (12) shows the relationship between the variables studied and the generation of these triglycerides. The coefficient of determination ($R^2$) for this model was 96.2% and Figure 9 shows the response surface graphs generated from Equation (12).

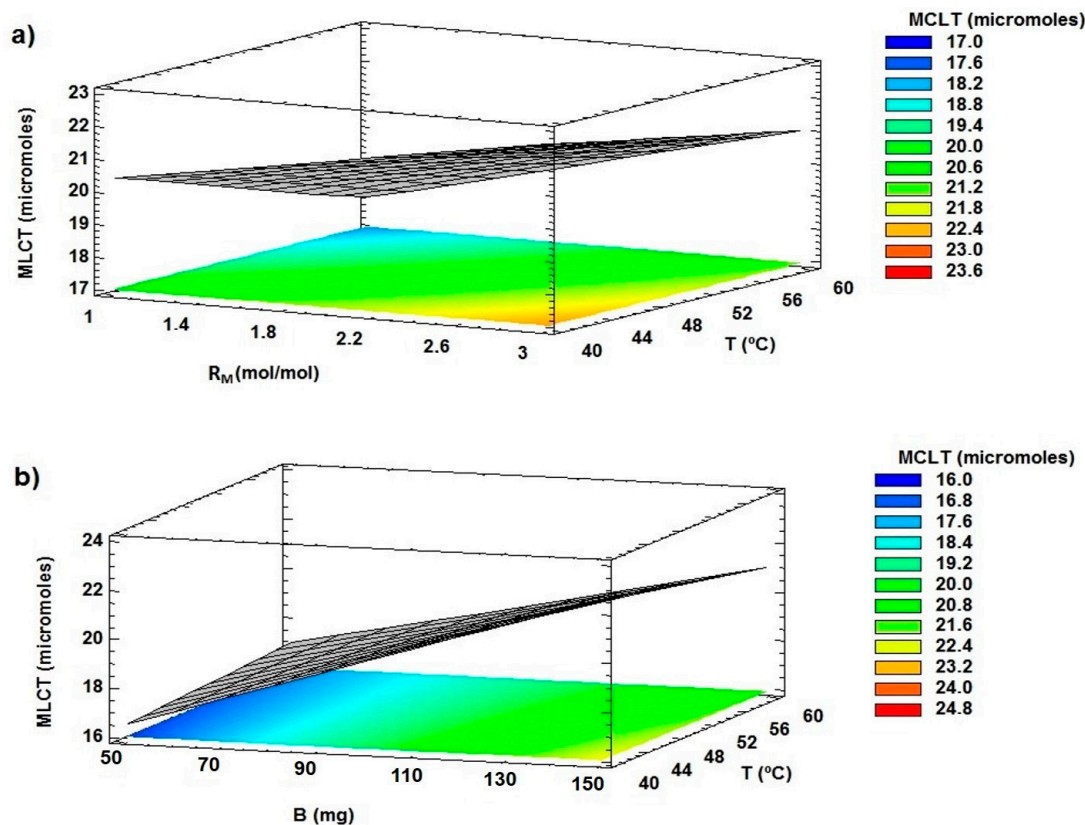

**Figure 9.** Generation of 1,3-dicaproyl-2-palmitoyl glycerol + 1,2-dicaproyl-3-palmitoyl glycerol (both classified as MLCT) by the esterification of dicaprin with palmitic acid catalyzed by BCL/chitosan. (**a**) Variables: $R_M$ and reaction temperature for a biocatalyst dosage of 150 mg, (**b**) Variables: mass of biocatalyst and reaction temperature carrying out the reaction with the highest proportion of palmitic acid evaluated in this study.

Products Generated from the Esterification of Dicaprin and Palmitic Acid

Several of the compounds present in the final reaction product are nutritionally interesting. MLM-type MLCT (1,3-dicaproyl-2-palmitoyl glycerol), MML-type MLCT (1,2-dicaproyl-3-palmitoyl glycerol) and MCT (tricaprin) are products with high added value. Further, 1,3-Diglycerides (1,3-dicaprin and 1-caproyl-3-palmitoyl glycerol) and 1,2-diglycerides with a medium-chain fatty acid at the sn-1/sn-3 position and a long-chain fatty acid at sn-2 (1-caproyl-2-palmitoyl glycerol) are also valuable glycerides.

Table 4 shows the percentage of all the nutritionally interesting acylglycerides present in this reaction system. The free fatty acids and glycerol were easily removed with an aqueous solution of KOH as previously reported [59]. In note "a" of Table 4, only the reaction products were considered. The final mixture was composed of between 46% and 66% of nutritionally interesting acylglycerides. The generation of nutritionally valuable acylglycerides was favored by the increase of the biocatalyst mass and the temperature. The concentration of palmitic acid did not have a statistically significant effect on this response. The relationship between the molar fraction of these acylglycerides (excluding 1,3-dicaprin) and the studied factors was appropriately adjusted by Equation (13) with $R^2 = 97.4\%$. A surface plot for this response is presented in Figure 10a, where the variation in the composition of the glycerides with nutritional value is shown as a function of the variation in the reaction conditions. It is evident that carrying out the reaction with high dosages of biocatalyst and with the lowest temperature favors the formation of medium- and long-chain triglycerides (MLCT) and medium- and long-chain diglycerides (MLCD).

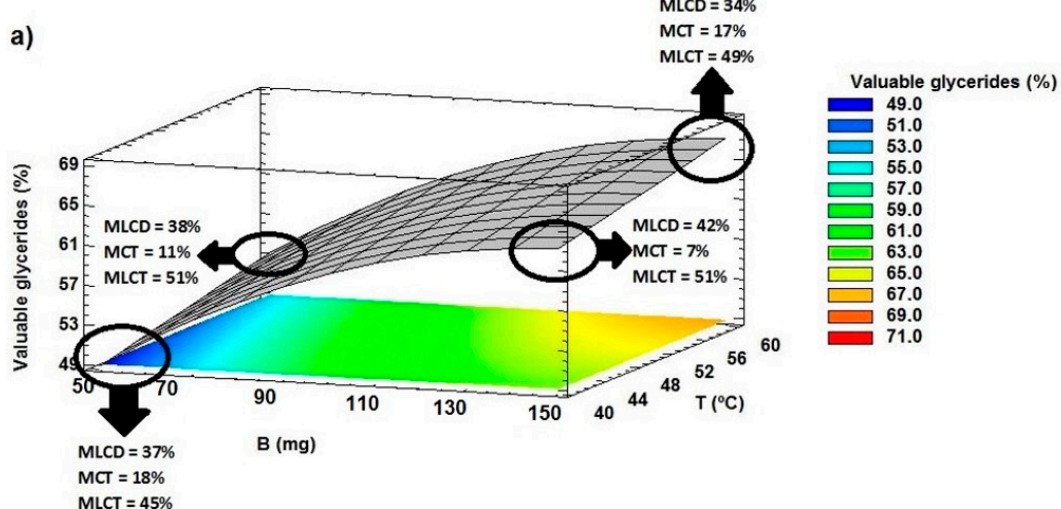

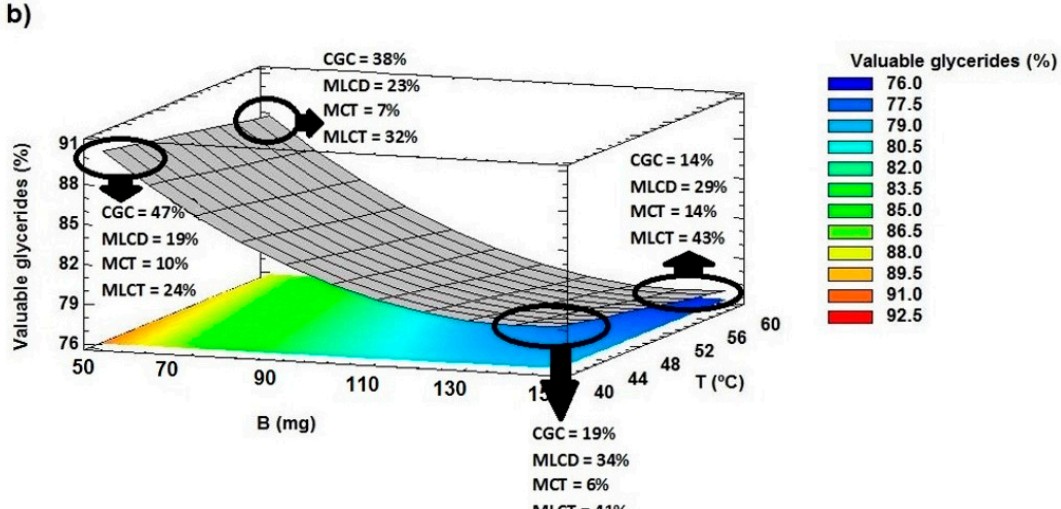

**Figure 10.** Synthesis of nutritionally valuable acylglycerides obtained from the esterification of dicaprin with palmitic acid catalyzed by the *Burkholderia cepacia* lipase immobilized on chitosan. (**a**) Molar fraction of the acylglycerides of interest not considering unreacted 1,3-dicaprin, variables: mass of biocatalyst and reaction temperature. (**b**) Molar fraction of acylglycerides in the final reaction mixture considering unconsumed 1,3-dicaprin, variables: mass of biocatalyst and reaction temperature for a $R_M$ of 3:1 mol/mol. CGC = 1,3-dicaprin, MLCD = medium- and long-chain diacylglycerides, MCT = medium-chain triacylglycerides, and MLCT = medium- and long-chain triacylglycerides.

The conversion of 1,3-dicaprin reached values between 50 and 90%. 1,3-Dicaprin is considered a compound of importance present in the final mixture. Note "b" of Table 4 shows the percentage of acylglycerides in the final product considering unreacted 1,3-dicaprin as a nutritionally valuable acylglyceride. In this case, the final blend consisted of 76–90% acylglycerides with high nutritional value (depending on the reaction conditions).

Equation (14) relates the fraction of nutritionally valuable glycerides to the studied factors. This equation was obtained from multiple linear regression, and only the statistically significant effects were considered. The coefficient of determination ($R^2$) for this model was 99.7%. In Figure 10b, the changes in the mixture of the glycerides of interest as a function of the reaction conditions are shown.

The synthesis of acylglycerol mixtures with different compositions (of high nutritional value) could be performed according to the requirements by modifying the reaction conditions.

## 3. Materials and Methods

### 3.1. Materials

Lipozyme RM IM, a commercial biocatalyst based on *Rhizomucor miehei* lipase immobilized on Duolite A-568, was kindly provided by Novo Nordisk A / S (Brazil). *Burkholderia cepacia* lipase diluted in dextrin, which is commercially available as a solid called Lipase PS "Amano", was generously donated by Amano Enzyme Inc. (Nagoya, Japan). Cicarelli Laboratories supplied silica gel, glycerol, and n-heptane. Capric acid, 1,2,4-butanetriol, silylation reagents, and tripalmitin were obtained from Fluka. Sigma-Aldrich provided monocaprin, dipalmitin, trilaurin, tricaprilin, tricaprin, trilaurin, trimiristin, and porcine pancreas lipase. Anedra S.A. supplied phenolphthalein and pyridine. Absolute ethanol was purchased from Dorwil and Primex S.A. (Iceland) provided chitosan. All products were analytical grade.

### 3.2. Adsorption of Glycerol on Silica Gel

In order to avoid glycerol inhibition, this polyol was adsorbed on silica gel following a previously reported methodology [54].

### 3.3. Enzymatic Esterification of Glycerol and Capric Acid

The esterification of glycerol and capric acid was carried out in 10 ml flasks and the reaction was catalyzed by Lipozyme RM IM. The reaction conditions were established by a previously reported experimental design [54].

The biocatalyst mass was added in two stages to minimize inhibition in the reaction medium: 50% of the biocatalyst was added at the beginning and the remaining 50% of biocatalyst after three hours of reaction. The analysis of the composition of the reaction samples was performed by gas chromatography (GC) with a procedure previously used for acylglycerides [12,54,57].

Table 1 shows the values of the experimental factors and the responses obtained for the factorial design $2^3$ with two central points created with the STATGRAPHICS Centurion version XV.2 software [54].

### 3.4. Immobilization of Burkholderia Cepacia Lipase

*Burkholderia cepacia* lipase (BCL) was immobilized on chitosan flakes. To perform the immobilization of the lipase, 500 mg of the Lipase PS "Amano" powder was placed in 50 mL of distilled water with pH = 6.5. The solution was stirred and then centrifuged at 8000 rpm for 5 min. The supernatant solution was recovered and contacted with 1 g of chitosan. The immobilization process was carried out with magnetic stirring at 400 rpm for 1 h and at 35 °C. The solid was recovered by filtration and dried at 30 °C for 15 h. The biocatalyst was placed in 50 mL of distilled water and magnetically stirred for 1 min to remove weakly adsorbed lipase. This washing procedure was performed in duplicate. Finally, the solid was dried in an oven at 30 °C for 15 h and then at 45 °C for 24 h. The biocatalyst obtained was called BCL/chitosan.

The degree of immobilization of the enzyme was determined by measuring sulfur by inductively coupled plasma atomic emission spectrometry (ICP-AES) [58]. The amount of sulfur is directly related to the mass of lipase present in the solution based on molecular weight [61] and amino acid sequence [62].

### 3.5. Dicaprin Esterification

Dicaprin obtained under the optimal conditions derived from the experimental design carried out as indicated in Section 3.3 was purified by a liquid–liquid extraction process developed by

Sánchez et al. [62]. The esterification of dicaprin and palmitic acid was performed in 10 ml vials. The vials were placed in thermostatics baths with temperature adjustment and the stirring was magnetic at 700 rpm. The esterification reaction was carried out as follows: 32 mg of dicaprin were dissolved in 2 ml of n-heptane and then the palmitic acid mass fixed for each study was added. The reaction began with the addition of the biocatalyst (BCL/chitosan) once the reaction mixture reached the temperature defined according to the experimental design described below. The choice of solvent was performed according to Bi et al. [63]. Solvents with a log P value higher than 4.0 increased the degree of esterification at the sn-2 position.

The values of temperature (T), palmitic acid/dicaprin molar ratio ($R_M$), and biocatalyst charge (B) were established according to the following experimental design.

The analysis of the composition of the reaction samples was performed by gas chromatography (GC) with a procedure previously used for acylglycerides [12,54,57,60]. In the case of structured triglycerides, GC analysis was combined with hydrolysis catalyzed by porcine pancreas lipase according to a previously reported methodology [12].

### 3.6. Experimental Factorial Design and Statistical Analysis

A factorial design $2^3$ with two central points was selected to carry out the study of the esterification of dicaprin and palmitic acid. A total of 10 experiments were performed. In this study, different responses were analyzed. Compounds consumed: 1,3-dicaprin (CGC), 1,2-dicaprin (CCG) and palmitic acid (P). Compounds generated: capric acid (C), glycerol (G), monopalmitin (PGG + GPG), 1-caproyl-2-palmitoyl glycerol + 1-caproyl-3-palmitoyl glycerol (CPG + CGP), tricaprin (CCC), dipalmitin (PPG + PGP), 1,3-dicaproyl-2-palmitoyl glycerol + 1,2-dicaproyl-3-palmitoyl glycerol (CPC + CCP) and 1-caproyl-2,3-palmitoyl glycerol (CPP). The production of acylglycerides with high nutritional value was also evaluated.

Tables 2–4 show the values of the experimental factors and the responses obtained for the experimental design. Both the design and the statistical analysis were performed using the STATGRAPHICS Centurion version XV.2 software. The response variables were adjusted by multiple regression and the models were refined by eliminating the variables without statistically significant effect using the F-test. The level of confidence of the adjustment was evaluated from the coefficient of determination ($R^2$). The ANOVA test was used to determine the statistical value of the variables.

## 4. Conclusions

The valorization of glycerol through the synthesis of acylglycerides with high nutritional value was carried out by a two-stage enzymatic process. In the first stage, glycerol was esterified with capric acid in a reaction catalyzed by Lipozyme RM IM. Under the optimal conditions, a conversion of 73% of capric acid was achieved with a selectivity of 76% to dicaprin. Of the total dicaprin generated, 93% corresponded to 1,3-dicaprin. The factors that reduce the production of the specific diglyceride were analyzed. Enzyme intervention in the acyl migration reaction could be possible. A simple theoretical model on lipase-mediated acyl migration was proposed and this would explain the experimental results.

In the second stage, the esterification of dicaprin with palmitic acid was catalyzed by the *Burkholderia cepacia* lipase immobilized on chitosan. A simple immobilization process generated an active biocatalyst for the esterification of the sn-2 position of 1,3-dicaprin. A mixture of acylglycerides consisting mainly of glycerol esterified with capric acid at sn-1 and sn-3 positions and with palmitic acid at sn-2 was obtained as the reaction product.

The reaction product was composed of acylglycerides with high nutritional value such as 1,3-dicaprin, 1-caproyl-3-palmitoyl glycerol, 1-caproyl-2-palmitoyl glycerol, 1,3-dicaproyl-2-palmitoyl glycerol, 1,2-dicaproyl-3-palmitoyl glycerol, and tricaprin. Depending on the reaction conditions, the fraction of valuable acylglycerides was between 76% and 90%.

The hydrolysis reaction favored the generation of unwanted acylglycerides. Controlling this reaction is essential to maximize the selectivity to the desired product. On the other hand, the presence of 1,2-dicaprin in the starting reagent favored the generation of products with lower nutritional value. Minimizing the concentration of this diglyceride is essential to maximize the nutritional value of the reaction product.

The variation in the reaction conditions in the second stage generated products with different acylglyceride compositions. This process could be implemented for the production of nutritional glycerides with composition as required.

**Author Contributions:** D.A.S., G.M.T, and M.L.F. planned the study, D.A.S. performed the experiment, processed the data, analyzed the data and wrote the manuscript, G.M.T., and M.L.F. revised the manuscript, M.L.F., and G.M.T. obtained the funds for this investigation. All authors have read and agreed to the published version of the manuscript.

**Funding:** This research was funded by the Agencia Nacional de Promoción Científica y Tecnológica (National Agency of Scientific and Technological Promotion, Argentina), grants PICT 2015-932/1583 and PICT CABBIO 2016-4687, and the Universidad Nacional del Sur, grant PGI 24/M141.

**Acknowledgments:** The authors thank the Agencia Nacional de Promoción Científica y Tecnológica (National Agency of Scientific and Technological Promotion, Argentina), the Consejo Nacional de Investigaciones Científicas y Técnicas (National Council for Scientific and Technological Research, CONICET) for the financial support and the Universidad Nacional del Sur.

**Conflicts of Interest:** The authors declare no conflict of interest.

## Abbreviations

BCL, *Burkholderia cepacia* lipase; C, Capric acid; CCC, Tricaprin; CCG, 1,2-dicaproyl glicerol; CCP, 1,2-dicaproyl-3-palmitoyl glicerol; CGC, 1,3-dicaproyl glicerol; CGP, 1-caproyl-3-palmitoyl glicerol; CPC, 1,3-dicaproyl-2-palmitoyl glicerol; CPG, 1-caproyl-2-palmitoyl glicerol; CPP, 1-caproyl-2,3-palmitoyl glicerol; DAG, Diacylglycerols; FA, Fatty acids; G, Glycerol; LCFA, Long chain fatty acids; MCFA, Medium chain fatty acids; MCT, Medium chain triacylglycerols; MLCD, Medium-long chain diacylglycerols; MLCT, Medium-long chain triacylglycerols; P, Palmitic acid; PGP, 1,3-dipalmitoyl glicerol; PPG, 1,2-dipalmitoyl glicerol; PPL, Porcine pancreas lipase; RML, *Rhizomucor miehei* lipase; TAG, Triacylglycerols.

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
