# Peer review of "Valorization of Glycerol through the Enzymatic Synthesis of Acylglycerides with High Nutritional Value"

_catalysts, doi:10.3390/catal10010116_

Round 1

Reviewer 1 Report

The paper describes the production of acylglycerides from the selective esterification of glycerol with an enzyme-based catalyst. With this strategy, the authors are able to obtain high added value products with high nutritional value.

Transformation of glycerol, the most important by-product from the biodiesel industry is important due to the high production in comparison with its lower demand for the food, animal feed and polymer industry. Its transformation into high added value products in order to contribute to circular economy concept is gaining interest due to increasing environmental concern in society.

In this paper, the enzymatic transformation of glycerol is described and the products and by-products generated are described and in somehow analysed and characterized; in addition, the mechanism for the acyl migration is proposed and the influence of biocatalyst loading, temperature and presence of glycerol in isomer formation is described.

The work is well presented in general and results are in accordance with text statements. However, in my opinion, to be on the scope of Catalysts, there is a lack of attention on the catalyst for this reaction:

In general, in the material and methods section, it would be desirable at least a brief description of the methodologies and procedures employed, not just refer the reader to the original method, which could be modified in order to adjust it to determined conditions. Catalyst preparation and performance must be emphasized. Catalyst composition analysis must be addressed to know enzyme loading and surface properties such us porosity and surface area. Also, as the enzyme has been immobilized in chitosan flakes, to be used in heterogeneous (as I understood) reusability should be addressed. The authors determined the influence of reaction parameters on isomer formation with a well-deduced proposal, If this has not been stated by any group before, must be mentioned; in the same way if this comes from the inspiration with other research works

With this corrections, the paper could be suitable for publication in Catalysts.

Author Response

We appreciate your comments and/or suggestions. We have added information on the immobilization methodology, support characterization, and immobilization efficiency. Since it is a first study of this biocatalyst in the esterification of 1,3-dicaprin, the reuse of the biocatalyst has not been evaluated yet. However, it is valuable information that will be taken into account in future work where we will seek to improve the efficiency of immobilization and to reduce the hydrolysis rate.

On the other hand, we have added a paragraph where we explain the origin of the proposed acyl migration model along with our and other authors' works that support it.

Reviewer 2 Report

The manuscript is very interesting and discusses important research problems. However  I recommend to describe with more details the conclusions section. The summary might be enriched with more research results.

Author Response

The authors appreciate your comment.

Both the conclusions section and the summary were expanded.

Reviewer 3 Report

The manuscript reports some novelty and interesting results. Unfortunately, the manuscript needs minor revision for several different reasons as outlined below.  

Abstract. “In this work, acylglycerides with high nutritional 14 value were obtained by enzymatic esterification in a two-stage process. In this work, nutritional and 15 medically interesting glycerides were obtained by enzymatic esterification through a two-stage 16 process.” These two sentences are repetitive. Please rephrase.

Lines  106-108 and 111-113. Please rephrase these sentences.

Immobilization of lipase from Burkholderia on chitosan. Essential immobilization parameters such as expressed activity and immobilization performance are missing.

Author Response

We welcome your comments and suggestions. They have been taken into account and the manuscript has been modified.

The description of the lipase immobilization methodology was extended and information about the characteristics of the support and the immobilization efficiency was added. On the other hand, enzymatic activity is usually determined in hydrolysis reactions (for example, triolein hydrolysis) or simple esterifications (for example, ethyl oleate synthesis) under specific conditions. We believe that the activity calculated in this way would not be representative of the activity of the biocatalyst in the reaction we have studied. The reaction is considerably more complex than those used to determine activity and the reaction conditions are variable. Finally, with the idea in mind of a process with the potential to be scaled, the biocatalyst will be measured and added in units of grams and not in term of Activity Units (AU).